# Predicting Breaststroke and Butterfly Stroke Results in Swimming Based on Olympics History

**DOI:** 10.3390/ijerph18126621

**Published:** 2021-06-20

**Authors:** Maciej Hołub, Arkadiusz Stanula, Jakub Baron, Wojciech Głyk, Thomas Rosemann, Beat Knechtle

**Affiliations:** 1Institute of Sport Sciences, Jerzy Kukuczka Academy of Physical Education, 40-065 Katowice, Poland; m.holub@awf.katowice.pl (M.H.); a.stanula@awf.katowice.pl (A.S.); j.baron@awf.katowice.pl (J.B.); w.glyk@awf.katowice.pl (W.G.); 2Institute of Primary Care, University of Zurich, 8091 Zurich, Switzerland; thomas.rosemann@usz.ch; 3Medbase St. Gallen Am Vadianplatz, 9000 St. Gallen, Switzerland

**Keywords:** swimming, athletic performance, breaststroke, butterfly stroke, analysis, Olympic Games

## Abstract

Here we describe historic variations in Olympic breaststroke and butterfly performance and predict swimming results for the 2021 Olympic Games in Tokyo. The results of the finalists, winners, and last participants in the women’s and men’s finals were analyzed, and a mathematical predictive model was created. The predicted times for the future Olympics were presented. Swimming performance among Olympians has been steadily improving, with record times of 18.51 s for female finalists in the 100 m butterfly (a 24.63% improvement) and 31.33 s for male finalists in the 200 m butterfly (21.44%). The results in all analyzed groups showed improvement in athletic performance, and the gap between the finalists has narrowed. Women Olympians’ performances have improved faster than men’s, reducing the gap between genders. We conclude that swimming performance among Olympians is continuing to improve.

## 1. Introduction

Swimming is a popular sport and is the second-largest based on the number of athletes at the Olympic Games [1]. This discipline has been included in the Olympic program since the first modern Games, which took place in 1896 in Athens, Greece. For over 100 years, swimming, as a sport, has been approached in different ways. At the beginning of the twentieth century, the discipline was shaped by amateurs. Nowadays, performance and professionalism are at such a high level that amateur athletes have little to no chance of winning medals and finals at international events [2]. Amateurs have been displaced by professionals who have devoted their lives to their dreams and accomplishments [3]. Competition results reflect this change because the level of performance has continually developed [4]. From decade to decade, athletic performance has been increasing, and old records are replaced, hoisted to limits that until recently had been unattainable [5]. This is due to improvements in human capabilities, the improved knowledge of sports coaches, scientific and sports advancement, and various technological advancements [6,7].

The breaststroke and butterfly stroke are symmetrical swimming techniques. They are the most technically and physically demanding of swimming strokes [8,9,10]. This is because of their low economy of movement, exhausting the swimmer more quickly [11]. A general profile of the energy expenditure of the four basic swimming styles shows that the symmetrical techniques—that is, breaststroke and butterfly stroke—are less economical than the asymmetrical—that is, front crawl and backstroke [11,12,13]. For this reason, the symmetrical techniques are limited to a distance of 200 m [14]. It is worth noting that the breaststroke was the first swimming technique, from which the others evolved, and breaststroke was dissociated from the butterfly stroke only in the 1950s [15].

The modern Olympic Games are part of a global culture and history, but most importantly, the most prestigious sports event in the world [16]. The Olympic Games is the most important sports event for many disciplines, including swimming [17]. A tendency has been observed, based on numerous studies, that a year before the Olympics, swimming performance improves and participants achieve better results in comparison with the Olympics in previous years [18,19]. There are several reasons for this, but this fact can be explained by the preparations for the Olympic trials and achieving the minimum qualifying times to participate at the Olympics. Coaches adjust their training plans to achieve the best results exactly in this most important sports event. The analysis and prediction of sports results in this work were based on the Olympic Games because they are the most prestigious and representative competition in this sport [20]. It should be noted that the organizers of the Tokyo Olympic Games in 2020 decided to postpone the event by 1 year due to the COVID-19 pandemic, which is an unprecedented step in modern history.

The general availability of sports results makes it possible to analyze variability in many different competitions and sports disciplines while observing results from previous years in a given period of time. This makes it possible to estimate and forecast future results [21]. Monitoring the progress and trends in swimming allows us to predict the direction in which this discipline is heading [4]. We can predict the stabilization or progression of performance, which affects the entire swimming community, especially training staff and the athletes themselves [18]. It also creates an opportunity to predict specific results important for sports associations and coaching staff [22,23]. Conducting studies to predict future sports results can be useful for developing appropriate training plans and strategies.

Here we investigate changes in athletic performance in the breaststroke and butterfly stroke over the course of the entire history of the Olympic Games. For this purpose, we developed a predictive mathematical model that estimated times for the future Olympics, which are to take place in Tokyo, Japan, in 2021. Assuming that athletic performance is constantly improving, we speculate what mechanisms may underlie these changes. We also wish to explore whether the inequality between men’s and women’s results are beginning to narrow. Based on related scientific articles, the results from this work will be compared with freestyle results to find similarities and differences, and an answer will be formulated to the question of how the development of the entire swimming discipline is being shaped.

## 2. Materials and Methods

### 2.1. Subjects

Results of men and women in the 100 m and 200 m breaststroke and butterfly stroke were collected from Internet sources (www.olympic.org, www.wikipedia.org, accessed on 6 April 2021). Six out of the eight competitions listed appeared for the first time in the Olympic program in the years 1956–1968. The oldest competition is the 200 m breaststroke, which made its debut in London in 1908 (men’s) and in Los Angeles in 1924 (women’s). We analyzed the time of the first and last participant in the finals and the mean time of the finalists (the most representative and reliable study group). Both swimming styles were considered one, which was due to the similar specification of limb movement and coordination, but, with the increase in athletic performance, the International Swimming Federation (FINA) dissociated them in 1952. As part of this work, an analysis of results achieved over the course of the entire history of the Olympics was conducted and was then applied to produce a predictive mathematical model, based on univariate linear regression analysis, to calculate the estimated times of the winners, finalists, and last participants in the Tokyo 2021 Olympic finals.

### 2.2. Statistical Analysis

Prognosticated times of swimmers in the Tokyo Olympics in 2021 were presented with an upper and a lower limit, with a 95% confidence interval. From this, an estimated value was computed. We employed a linear analysis of the regression of times in the men’s and women’s breaststroke and butterfly stroke (100 m and 200 m distances), which was dictated by the assessment of the rectilinear relationship between the years 1972 and 2016, as well as by a high Pearson linear correlation. We detected an extremely strong relationship between the year of the Olympics and the times achieved in various distances. The additional advantage of employing univariate models was the presence of a normal distribution in times achieved in all swimming distances (Shapiro–Wilk test, statistical significance). A lack of extreme values helped to fit the regression line accurately to the data. The univariate analysis of variance confirmed that the models built accurately fit the data. The coefficient of determination for all analyses reached high values, which explained the variability of times in the distances analyzed by as much as 96%.

## 3. Results

### 3.1. 100 m Breaststroke

Figure 1 reveals a growing tendency to improve performance in men’s and women’s results. The only exception is a regression in men’s results in the Moscow Olympics in 1980, by 0.30%, and in women’s results 12 years later in Barcelona, by 0.27%. Significant progressions were made in the early history of this event, then less so. The most substantial progress was achieved in 1972 (Munich) and 1976 (Montreal) by men and in 1976 (Montreal) and 1980 (Moscow) by women; 3.63% and 2.61%, and 3.48% and 2.68%, respectively. It is worth noting that the last several editions of the Olympics yielded a significant progression by men: in Beijing in 2008, by as much as 1.51 s (2.47%), and by women 4 years later in London, by 1.65 s (0.96%). These are substantial differences by today’s standards, in which distance and style specializations are on a much higher level than several dozen years ago. In 48 years, men’s finalists have improved by 9.41 s (13.76%) and women’s by 10.05 s (13.14%). These results indicate that both sexes have experienced significant improvements in athletic performance, but without a meaningful difference between each other, although the slightly steeper progression of times in women’s is worth noting. Additionally, the difference among the winners of the Olympics in Rio (2016) was 7.80 s. This is the highest result in the last 32 years, namely, since the Olympics in Los Angeles (1984). Such an observation upsets the notion that women’s and men’s results are converging. This, however, is an individual case, a phenomenal performance by Adam Peaty, who achieved one of the most spectacular results in the history of swimming.

In Table 1 we predict the results for the upcoming Olympics. The Pearson correlation coefficient in men was 0.98, which meant a strong link between the Olympics and the time at that distance, while a coefficient of determination of 96% explained the time in the distance analyzed. In women, these coefficients were 0.95 and 89%, respectively. Based on these data, the men’s results were more consistent than women’s. The probable times were estimated to be 0:58.12 for the men’s finalists and 1:04.44 for the women’s finalists. It is worth paying attention that, throughout history, only one man and three women swam better results than these.

Figure 2 shows an almost constant improvement in results since the beginning of the twentieth century. In the men’s competition, two periods stand out with high result progression values: between the Paris (1924) and Helsinki (1952) Olympics and between the Munich (1972) and Montreal (1976) Olympics. For the women’s competition, one long period of significant progress was observed between the Amsterdam (1928) and Moscow (1980) Olympics. Because technical conditions varied widely prior to 1952, especially from the subsequent editions, this analysis will begin from this time.

Men and women displayed a clear tendency to progress in athletic performance. The most significant progressions were achieved in 1952 (4.31%), 1964 (4.98%), and 1972 (4.26%) among men, and in 1952 (2.98%), 1972 (2.95%), and 1972 (3.88%) among women. For the men, a decrease in performance was noted at three events, with the greatest decrease in Rome (1960) (by 1.49 s, 0.94%). In women, a decrease in performance occurred at two events, of which the largest was in Barcelona (1992) (by 0.49 s, 0.29%). The total progression of men’s results was 29.69 s (18.85%), and that of women’s was 35.37 s (19.90%). These findings clearly indicate that the most significant progress was achieved by women, simultaneously decreasing the difference between the sexes. This is also evident by the fact that the gap between the gold medalists in 1952 (Montreal) was 17.30 s and 4.46 s less in 2016 (Rio), reaching 12.84 s.

Using predictive mathematical models, we expect the men’s finalists to achieve an average time of 2:05.10 in the 200-m breaststroke in the Tokyo Olympics in 2021, with the women’s finalists achieving a mean time of 2:17.99. These times might be exaggerated, despite a highly accurate fitting of the models, especially among men (r = 0.97; r^2^ = 0.94). This conclusion comes to light after analyzing the world rankings and records. In men, the best result in history was exactly the same as that suggested by the predictive model for the eighth result in the Tokyo Olympics in 2021 (2:06.12; r = 0.97; r^2^ = 0.95). As a result, all finalists would finish the distance in less than the world record. This is, of course, possible, all the more so in such a dynamically growing distance, but still incredibly difficult to believe.

### 3.2. 100 m Butterfly

For the 100-m butterfly stroke (Figure 3), we detected a trend for improving results in both sexes. In early women’s results, the trend line resembles a wave, which then flattens in 1980 (Moscow). In the men’s results, the line is flat across the entire history, which leads to the conclusion that the results were more consistent. Women began competing in this distance 12 years earlier (1952) than men (1968). These first and several subsequent editions of the Olympics (held between 1960 and 1976) yielded an immense progression, by as much as 6.69% (4.83 s) in the Tokyo Olympics in 1964. Men experienced a significantly smaller progression, at most reaching 2.93% in Munich (1972), but only two times a result regression, namely, in Barcelona (1992) by 0.40% and in London (2012) by 0.88%. For women, a result regression occurred three times, in Moscow (1980), Atlanta (1996), and Athens (2004) (by 0.83%, 0.49%, and 0.11%, respectively).

The total result progression between the first and the last edition of the Olympics was 6.68 s (11.53%) for men. For women, this difference was greater: 18.51 s (24.63%). These results show how great of an improvement women have accomplished in 60 years and how close their results have approached men’s. This is proven by the fact that the gap between the gold medalists of the first Olympics hosting these competitions (Mexico 1968) and the last (Rio 2016) shrank from 9.60 s to 5.09 s, and the time gap in Tokyo (2021) is predicted to be only 4.81 s (49.85–54.66 s). According to this prediction of results for the Tokyo Olympics in 2021, there is a strong probability (r = <0.97, 0.98>; r^2^ = <0.95, 0.96>) that the men’s finalists will achieve an average time of 50.26 s, while women’s finalists achieve an average time of 55.78 s.

### 3.3. 200 m Butterfly

The 200-m butterfly stroke, considered one of the most difficult swimming distances, has also shown progression (Figure 4). Since the beginning of this competition in 1980 (Moscow) and 1988 (Seoul), for women and men, respectively], performances have improved. The most marked improvement in athletic performance for women was in Munich (1972) and Montreal (1976) (by 6.58% and 4.00%) and for men, in Rome (1960) and Munich (1972) (by 6.42% and 5.14%, respectively). Across the 13 editions of the Olympics, women experienced a decrease in performance only two times: in 2004 (Athens) and 2016 (Rio), by 0.31% and 0.19%, respectively. Men experienced a decrease in performance four times: in 1980 (Moscow), 1988 (Seoul), 2012 (London), and 2016 (Rio), by 0.81%, 0.19%, 0.34%, and 0.47%, respectively. Interestingly, men showed a regression in the last two events. These are small values, albeit visibly upsetting a trend of almost constant progression. The general tendency on the graph reveals a decreasing gap between finalists. The widest was for women in 1984 (Los Angeles), by as much as 10.49 s. In the last two editions, in 2012 (London) and 2016 (Rio), this parameter was 3.36 s on average, with a visible downward tendency. Differences between women and men show a downward tendency, although they have been less pronounced in the last several editions. The general progression of women’s results was 14.93% (22.19 s), and that of men’s was 12.17% (15.90 s), counting from 1968 (Mexico). This is another competition in which women have made an impressive improvement.

Predictions for the Tokyo Olympics in 2021 reveal a clear progression. Average times for the women’s and men’s finalists are expected to be 2:04.02 and 1:52.40, respectively. In comparison with results from the last Olympics in 2016 (Rio), women will improve their results by 2.38% and men by 2.37%. Both of these values are similar but, most importantly, significantly high. The most marked progression, according to the predictive model, will be achieved by the eighth participant in the men’s finals: by as much as 2.90% (3.40 s).

The result trend line between women and men has evidently fattened, decreasing the differences that were clear just several dozen years ago. However, the prediction of results based on mathematical models shows a clear tendency toward important changes in the Tokyo Olympics (2021). Table 2 presents the results of the winners in Rio (2016) and the results predicted for the next edition. All participants starting in the eight competitions will achieve a result progression, simultaneously decreasing differences between women and men by as much as 2.33% in the 100 m breaststroke.

## 4. Discussion

The analysis presented in this work reveals a clear increase in athletic performance across all distances in the women’s and men’s breaststroke and butterfly stroke in the Olympic Games. Comparing both sexes, a converging tendency is detected, which is a consequence of women’s higher result progression. The reasons for the women’s results improving are manifold. Men began to participate in professional sport earlier than women, which was dictated by historical, social, and cultural circumstances [24]. Women faced difficulties in their roles in sport being accepted but were eventually able to overcome exclusion and stereotyping, which discriminated against them and hampered their growth. Girls were not involved in sports as much as boys, deepening the differences early in life and strengthening the notion of “masculine sport” [25]. As a consequence, a separation of disciplines by sex came about [26], clearly divesting women of opportunities for growth. Although some stereotypes have survived to this day, sportswomen are winning the battle for women’s rights and respect, which translates into greater participation in sports competitions.

Social and cultural factors had a significant impact not only on women’s performance but also on men’s. There are many examples, but I will give two that arouse controversies. The first example will be the participation of Chinese swimmers at the Olympic Games. In the 20th century, China was not successful internationally and registered a small number of competitors to compete in the Olympics [27]. Everything changed at the beginning of the new century, when swimming in China became a beneficiary of the Chinese government committing financial and scientific support [28]. Since then, China has become a swimming power in the world.

The second example of the influence of social and cultural factors that shaped the reality of swimming was the infamous doping program in the GDR. The practice took place in the 1970 and 1980s. The communist government secretly introduced a national doping program that involved the forced drugging of athletes, thereby increasing their athletic abilities [29]. The aim of the program was to increase the popularity and prestige of the state. On a performance level, the system was successful [27]. East German athletes illegally achieved great success. The truth came out in the 1990s, but unfortunately, it was impossible to turn back time and alter the results and records.

The differences between the sexes in the economic profile have an impact on the sports results in swimming. Women’s swimming is characterized by a higher stroke rate and shorter stroke length than men, resulting in poorer performance [30]. Moreover, men have a higher stroke index than women [31]. These are a result of natural factors and predispositions, such as height, muscle mass, and hormone levels, which impact capacities such as strength, speed, jumping abilities, and endurance to varying extents [32]. However, women exhibit greater fatigue resistance than males [33], which is confirmed by research studies on sexual dimorphism. Research confirms the trend that the longer the effort, the smaller the disproportions between women and men will be, and the shorter and more intensive, the wider the divergence will be [34]. Analysis of the finalists confirms this tendency.

Mathematical models define trends based on prior observations. They predict future results but are not to be trusted blindly [35]. Strictly mathematical reasoning would lead to absurd results, including women excelling men and a progression to time of almost zero [36]. The predicted women’s result is driven by the trend that women are now progressing faster than men, which might be due to their having entered top-ranking sports competitions at a later time and not having achieved as advanced a level of growth [37].

This work showed a significant progression in the distances subject to analysis. Other swimming competitions not addressed here also show a continuous result progression. It is a discipline that is growing dynamically and is breaking more and more barriers. Professionalism is so high that swimmers focus mainly on a single, most important event, despite the multitude of competitions to choose from. This is because participants in different distances exhibit different capabilities. This way, advancement and specialization are growing, which impacts the athletic level of a given competition and the difficulty of combining several different starts [18,38]. Guiding a participant in a direction that would be the most appropriate requires careful monitoring [39,40]. Reasons for the constant improvement in swimming results are many-fold. One reason seems to be technological growth, which has led to numerous discoveries, such as superfast starting swimsuits [41]. Another reason is the growth of sciences, including medical sciences, which have been progressively more often involved in sport [6,7], including medicine, physiotherapy, biomechanics, biochemistry, physiology, and psychology. Even without these, swimmers’ morphological parameters have changed, leading to better results [42,43], as well as swimmers’ age, which has indeed increased [44]. Because of the growth of pharmacology, sports associations have taken an interest in working with doctors. This has led to the rise of illegal doping, in which sportspeople use drugs designed to treat the sick to improve their psychophysical capabilities [45]. As Heazlewood (2006) notes, sports results will also be influenced by changes associated with biomechanics and the technique of swimming, improved training programs, a larger population of sportspeople, and advanced programs identifying young sportspeople with predispositions to become champions. Economic factors will also be important, including wages, sponsorship agreements, and other financial profits, which are now incomparably higher than in the past.

In summary, it is clear that there has been a constant progression in athletic performance among professional swimmers. Comparing the results from this work with a related scientific article on the freestyle stroke in swimming, the conclusions are similar. As Stanula (2012) notes, “As both male and female athletes tend to compete more and more vigorously within their groups, the gap between the gold medalist and the last finisher in the final is constantly decreasing, which provides the best evidence that this sport discipline continues to develop”. Among the two genders, women demonstrated a more significant development of performance, which brought their results closer to those of the men. The results calculated for both sexes for the future Olympic Games also show a substantial progression in athletic performance compared to the prior event. Numerous factors play a role here, but stagnation in athletic performance is expected to happen eventually, when the absolute peak of human capabilities is reached. We are unable to determine exactly when this will occur, but according to the results of this analysis, it will not be in the near future. The impact of the covid-19 pandemic on the results at the Olympics will not be investigated until the end of the event. We can speculate whether it will be positive or negative from scientific articles [46,47,48], but we cannot be sure.

## 5. Conclusions

Performance in professional competitive swimming has been steadily improving. In our analyses, all analyzed groups show a tendency to improve. This is evident from the results of the winners and finalists, but also from the times of the last participants in the finals and the time gaps among the medalists and finalists, which are decreasing.

## Figures and Tables

**Figure 1 ijerph-18-06621-f001:**
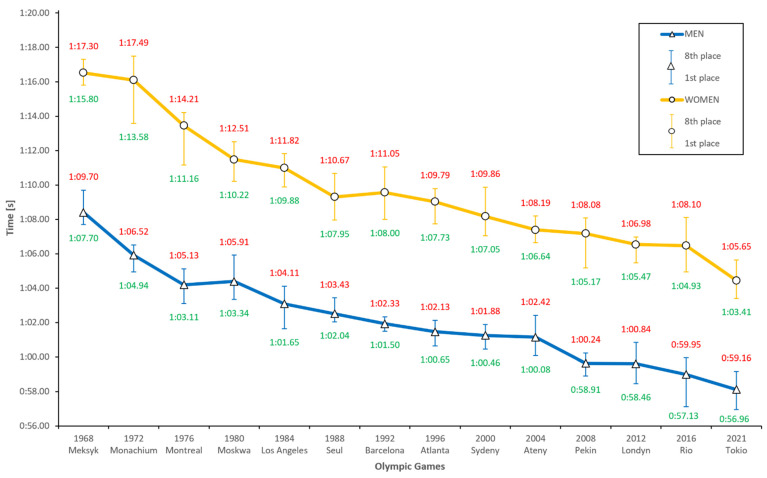
Men’s and Women’s 100 m breaststroke results during the last 13 Olympic Games and prediction for Tokyo 2021.

**Figure 2 ijerph-18-06621-f002:**
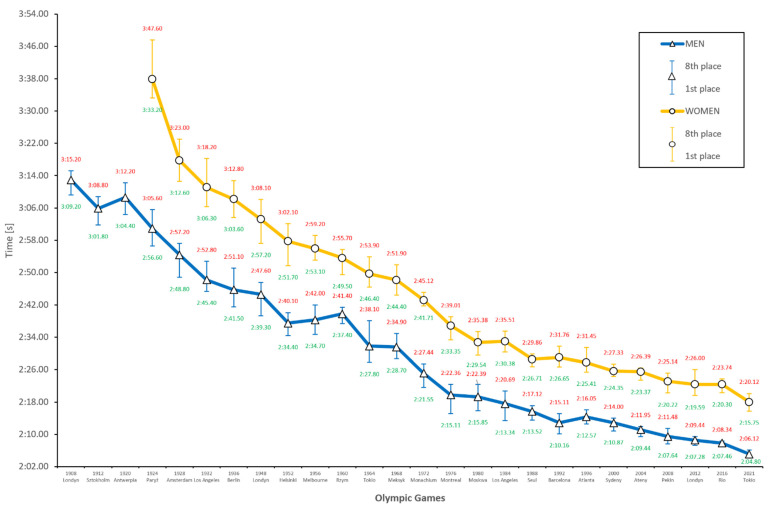
Men’s and women’s 200 m breaststroke results during the last 25 Olympic Games and prediction for Tokyo 2021.

**Figure 3 ijerph-18-06621-f003:**
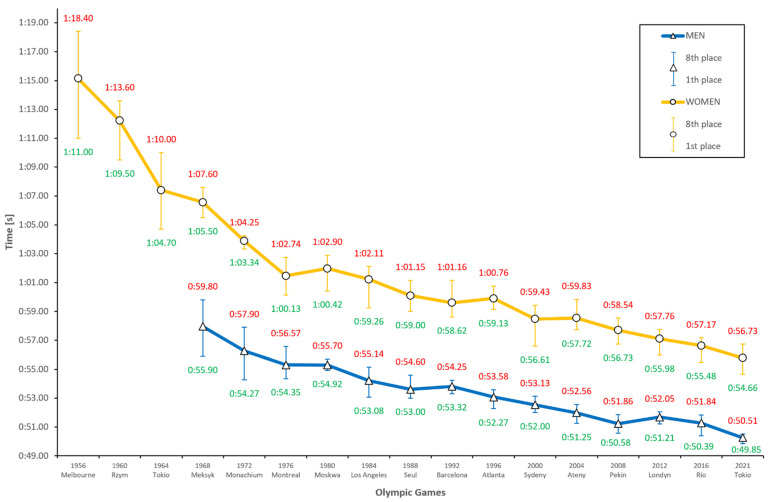
Men’s and women’s 100 m butterfly results during the last 16 Olympic Games and prediction for Tokyo 2021.

**Figure 4 ijerph-18-06621-f004:**
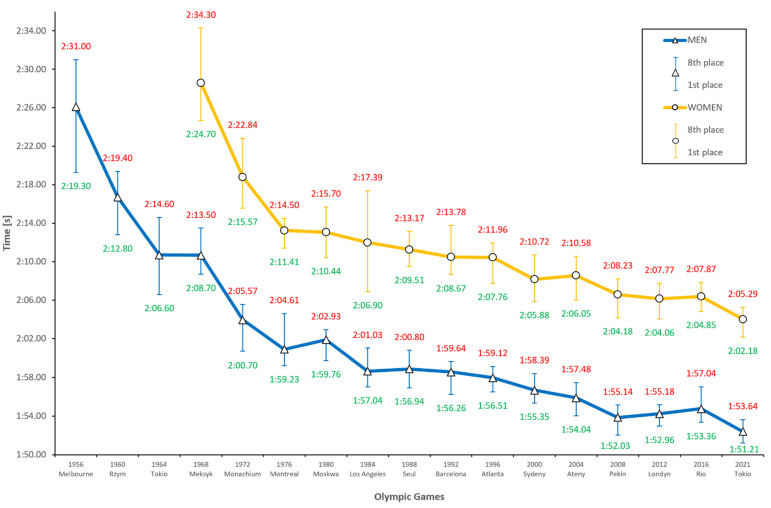
Men’s and women’s 200 m butterfly results during the last 16 Olympic Games and prediction for Tokyo 2021.

**Table 1 ijerph-18-06621-t001:** Summary of univariate linear regression for the results of the first and eighth place in the finals and the average time of the women’s and men’s finalists in the 100-m and 200-m breaststroke and butterfly stroke along with a prediction of times for the Tokyo Olympic Games in 2021.

Stroke	Distance	Sex	Place	Values of the Univariate Linear Regression Model				Prediction of the Results for the 2021 Olympics		
				F	df	r	r^2^	Prediction Value	Confidence Interval 95%	
									LL	UL
Breaststroke	100 m	Men	1st	215.15 *	1. 10	0.98	0.96	0:56.96	0:56.27	0:57.65
			final	256.49 *	1. 10	0.98	0.96	0:58.12	0:57.52	0:58.73
			8th	113.13 *	1. 9	0.96	0.93	0:59.16	0:58.32	0:59.99
		Women	1st	129.77 *	1. 10	0.96	0.93	1:03.41	1:02.37	1:04.45
			final	83.96 *	1. 10	0.95	0.89	1:04.44	1:03.02	1:05.86
			8th	57.31 *	1. 9	0.93	0.86	1:05.65	1:03.87	1:07.43
	200 m	Men	1st	60.57 *	1. 10	0.93	0.86	2:04.80	2:02.46	2:07.14
			final	156.49 *	1. 10	0.97	0.94	2:05.10	2:03.21	2:06.99
			8th	144.26 *	1. 8	0.97	0.95	2:06.12	2:03.87	2:08.38
		Women	1st	63.86 *	1. 10	0.93	0.86	2:15.75	2:12.30	2:19.21
			final	82.89 *	1. 10	0.94	0.89	2:17.99	2:14.91	2:21.06
			8th	77.32 *	1. 10	0.94	0.87	2:20.12	2:16.92	2:23.33
Butterfly	100 m	Men	1st	96.48 *	1. 10	0.95	0.91	0:49.85	0:49.16	0:50.54
			final	231.19 *	1. 10	0.98	0.96	0:50.26	0:49.75	0:50.77
			8th	216.85 *	1. 10	0.98	0.96	0:50.51	0:49.90	0:51.12
		Women	1st	69.59 *	1. 10	0.94	0.87	0:54.66	0:53.50	0:55.83
			final	186.93 *	1. 10	0.97	0.95	0:55.78	0:55.06	0:56.50
			8th	236.34 *	1. 9	0.98	0.96	0:56.73	0:56.06	0:57.41
	200 m	Men	1st	109.31 *	1. 10	0.96	0.92	1:51.21	1:50.05	1:52.40
			final	107.46 *	1. 10	0.96	0.91	1:52.40	1:51.04	1:53.76
			8th	114.65 *	1. 10	0.96	0.92	1:53.64	1:52.21	1:55.07
		Women	1st	44.95 *	1. 10	0.9	0.82	2:02.18	2:00.03	2:04.33
			final	77.93 *	1. 10	0.94	0.87	2:04.02	2:02.20	2:05.84
			8th	45.89 *	1. 10	0.91	0.82	2:05.29	2:02.52	2:08.06

Note: The analysis involved results achieved in the Olympics since 1972 (N = 12). F—value of one-way analysis of variance, df—degrees of freedom, r—Pearson correlation coefficient, r^2^—coefficient of determination, LL—lower limit, UL—upper limit. *—*p* < 0.001.3.2. 200 m Breaststroke.

**Table 2 ijerph-18-06621-t002:** Results and differences for the winners in the Rio Olympics in 2016 and the results predicted for the Olympics in 2021.

Event		OG	Women	Men	Difference [s]	Difference [%]
Breaststroke	100 m	Rio 2016	1:04.93	0:57.13	7.80	13.65
		Tokyo 2021	1:03.41	0:56.96	6.45	11.32
	200 m	Rio 2016	2:20.30	2:07.46	12.84	10.07
		Tokyo 2021	2:15.75	2:04.80	10.95	8.77
Butterfly	100 m	Rio 2016	0:55.48	0:50.39	05.09	10.10
		Tokyo 2021	0:54.66	0:49.85	04.81	9.65
	200 m	Rio 2016	2:04.85	1:53.36	11.49	10.14
		Tokyo 2021	2:02.18	1:51.21	10.97	9.86

## Data Availability

Publicly available datasets were analyzed in this study. The data can be found here: https://olympics.org/en/; https://en.wikipedia.org (accessed on 25 March 2021).

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
