# Peer review of "Predicting Breaststroke and Butterfly Stroke Results in Swimming Based on Olympics History"

_ijerph, 2021, doi:10.3390/ijerph18126621_

Round 1

Reviewer 1 Report

I strongly believe that this analysis should have been done with a polynomial adjustment. I think that applying a linear adjustment is not appropriate or even real, in a context of high sporting performance in which it may be more and more difficult to achieve the prediction of which the authors speak, and therefore, the differences between some times and others are going to be increasingly narrower. During the first confinement caused by the pandemic, I used to play with data sheets in which I included the number of cases that my country reported each day. Through this prediction I was able to sense that the so-called "first wave" was going to last longer than our politicians had predicted because the trend line showed me daily that infections sometimes dropped at a faster rate, but at other times at a slower rate. I remember experimenting with predictions based on the different types of trend line: when the trend line was polynomial, it was sensitive to detecting the rate of infection, that is, whether each day there were more or less infections than the previous one; however, when the extrapolation was linear, the predictions pointed to the fact that infections were going to reach infinity, or that on the contrary, they were going to fall suddenly in a matter of several days; therefore, the polynomial adjustment was much more sensitive and accurate than the linear one.

I find the explanation and reasons for not using the pre-1972 data respectable because of the lack of consistency and quality of the marks, although perhaps that is not the subject of discussion here. I think that the authors should be aware and much more critical of the fact that it is the times achieved before 1972 that indicate that this adjustment cannot be made in a linear way, as they show that a long time ago, the improvement in times was much more pronounced than it is now, due to the lack of professionalisation and inequality between some competitors and others. Today, however, this is no longer the case, so I do not see why the authors do not want to take this into account in their analysis. In other words, a polynomial adjustment would be able to detect that the rate at which times are being reduced is slowing down, whereas a linear adjustment is not able to detect this effect. In fact, I would even appreciate to have seen the prediction of both extrapolations (linear and polynomial) and for the authors to have discussed the possible advantages or disadvantages of each adjustment.

This is my view, but of course it may be wrong, so it is up to the authors to consider whether to go ahead with the possible changes suggested. For the sake of transparency only and to avoid the possibility of the journal rejecting the paper in this step, I will mark minor revisions.

I will be happy to look at the article again if changes are made.

Good luck.

Author Response

Response to Reviewer 1 Comments

Answer: Thank you very much for all your comments and suggestions. We are very grateful for precise indication of the changes.

 We respect the polynomial regression suggestion very much, but unfortunately it is not our goal. We have seen a linear trend since 1972 and this is what we want to continue in our work. Please respect our decision regarding the linear model and thank you for the suggestion of a possible change. We appreciate it very much. In defense of our idea, I would like to add that statistical indicators such as the coefficient of determination and the Pearson correlation coefficient are very high and well suited to the data, which gives a reason to make a decision about the model. We chose linear analysis because there are strong reasons for this, which we described in the material and methods chapter.

We have made a list of the coefficients of determination of linear and polynomial regression and compared them in the table below. We respectfully agree that polynomial regression showed a better fit, but this is minimal and does not give rise to strong reasons to alter the entire work. The avarage of the coefficients of determination in linear regression is 0.91 and in polynomial regression is 0.94. Both are great matched and differ only by 0.03. The prediction of results is always subject to a certain amount of uncertainty, but the values of the determination coefficients at the level of over 0.90 have a solid basis for their use.

Thank you once again for any suggestions and comments, which we respect very much. We hope that the answers will satisfy.

Tab. List of coefficients of determination in linear and polynomial regression

STROKE

EVENT

SEX

PLACE

LINEAR

POLYNOMIAL

BREASTSTROKE

100 M

♀

8th

0,86

0,94

final

0,89

0,97

1st

0,93

0,97

♂

8th

0,93

0,93

final

0,96

0,97

1st

0,96

0,96

200 M

♀

8th

0,87

0,94

final

0,89

0,96

1st

0,86

0,93

♂

8th

0,95

0,97

final

0,94

0,96

1st

0,86

0,89

BUTTERFLY

100 M

♀

8th

0,96

0,97

final

0,95

0,95

1st

0,87

0,89

♂

8th

0,96

0,99

final

0,96

0,97

1st

0,91

0,91

200 M

♀

8th

0,82

0,85

final

0,87

0,92

1st

0,82

0,87

♂

8th

0,92

0,95

final

0,91

0,93

1st

0,92

0,93

AVARAGE

0,91

0,94

Reviewer 2 Report

This article analyses the male and female performances of breaststroke and butterfly swimmers to create a mathematical prediction model for the Tokyo 2021 Olympic Games.

Here are my contributions:

Introduction

The introduction is appropriate. It provides the necessary information and is written in an orderly and coherent way.

Materials and Methods:

Line 46 and 91, in which year were the styles dissociated?

Have the marks obtained with the latest generation of swimming suits been included?

Results:

Line 203 opens [, but does not close afterwards

I cannot find any reference in the text to table 1.

Discussion:

Have you considered the possible effect on the results of the pandemic suffered by the COVID-19? The discussion lacks any comment on the effect that the pandemic may have on the swimmers' training or modification of their sport planning, and its influence on the expected results. Is there any other similar moment in the history of the Olympic Games?

Author Response

Response to Reviewer 2 Comments

Thank you very much for all your comments and suggestions that helped me a lot to review the article again. I am very grateful for precise indication of the changes. Below You will find a point-by-point reply to all Your comments. All the changes have been highlighted in the attached revised version of the manuscript.

Introduction

The introduction is appropriate. It provides the necessary information and is written in an orderly and coherent way.

Materials and Methods:

Answer: Thank You for the comment, we have modified the text according to the reviewer’s instructions.

Line 46 and 91, in which year were the styles dissociated?

Answer: „Both swimming styles were considered one, which was due to the similar specification of limb movement and coordination, but, with the increase in athletic performance, the International Swimming Federation (FINA) dissociated them in 1952.”

Have the marks obtained with the latest generation of swimming suits been included?

Answer: „Reasons for the constant improvement in swimming results are manyfold. One reason seems to be technological growth, which has led to numerous discoveries, such as superfast starting swimsuits [41].”

Results:

Line 203 opens [, but does not close afterwards

Answer: Thank You for the comment, we have made the corrections.

I cannot find any reference in the text to table 1.

Answer: Thank You for the comment, we have modified the text according to the reviewer’s instructions.

„In Table 1 we predict the results for the upcoming Olympics. The Pearson correlation coefficient in men was 0.98, which meant a strong link between the Olympics and the time at that distance, while a coefficient of determination of 96% explained the time in the distance analyzed.”

Discussion:

Have you considered the possible effect on the results of the pandemic suffered by the COVID-19? The discussion lacks any comment on the effect that the pandemic may have on the swimmers' training or modification of their sport planning, and its influence on the expected results. Is there any other similar moment in the history of the Olympic Games?

Answer: Thank You for the comment, we have modified the text according to the reviewer’s instructions.

Introduction:

„It should be noted that the organizers of the Tokyo Olympic Games in 2020 decided to postpone the event by 1 year due to the COVID-19 pandemic, which is an unprecedented step in modern history.”

Disscusion :
„The impact of the covid-19 pandemic on the results at the Olympics will not be investigated until the end of the event. We can speculate whether it will be positive or negative from scientific articles [46–48], but we cannot be sure.”

This manuscript is a resubmission of an earlier submission. The following is a list of the peer review reports and author responses from that submission.

Round 1

Reviewer 1 Report

General comments

I read this work with interest and authors should be valorised by their work. However, the text present relevant structure, language and conceptual gaps that significantly decreased its overall quality. Some examples are given under these lines, but an extensive review is necessary. In addition, there are some vague and also some speculative sentences. Lastly, authors seem to minimize the importance of the differences between male and female swimmers regarding their specific economy profiles). So, while the research question is an interesting and potentially impactful one, the manuscript contains several shortcomings. This could be overlapped with the consulting of relevant bibliography that was not used and focusing on the essential (forgetting the accessory).

Specific comments

. There are very long sentences with redundant writing. Please focus on the essential. E.g.: title, first sentence of the abstract, saying that the winners are those in 1st place and so on and so forth. As an example, the expression “The results presented in the study show a tendency towards…” can be written “Data presented a tendency towards…”.

. Some sentences/expressions are hard to understand. E.g. “on their basis”, “The results in all analyzed groups showed improvement”, “The article verifies”, “ Competition results reflect this change “. There are many other examples, but I will not display more.

. Please include some numerical results at the abstract as this section should give a clear idea of the data obtained.

. “ swimming is one of the 20 sports that is constantly developing and making progress in terms of performance”. Were other sports analysed in the current study? If not, how can they be compared with swimming?

. Swimming is a sport or a discipline? Please be coherent.

. In L32-3 authors use the word “Nowadays” but add a reference from 2011. In addition, they say that “amateur sport has completely been removed from international event” but some data news to be shown and there are some international level events in which many swimmers are not professional.

. L36/7: I wonder why coaches knowledge that allow more efficient training processes is not referred here.

. L40: ref 8 is only about breaststroke. There is a nice paper of Barbosa et al (2008 J Sport Sc Med) and other from de Jesus et al J Appl Biomech 2012 about butterfly technical characterization that you can use here (and along the text).

. L41: I think you wanted to use ref 10 here instead of ref 9, no?

. Please notice that “a style” is the individual expression of a conventional technique. Thus, breaststroke and butterfly are swimming techniques, not swimming styles as erroneously disseminated.

. Freestyle is an event, not a swimming technique (please replace by front crawl).

. L45-7: please make some evidence about this speculative sentence.

. Please explain why “a year before the Olympics, swimming performance improves” by referring to the trials and selection programs swimmers need to attend and, also, to the A/B minimal times that are required to participate at the Olympics.

. “swimmers adjust their training plans”? To my knowledge the responsible for the training process are the coaches not the swimmers.

. L60/1: this sentence does not make any sense to me.

. L62: this sentence does not make any sense to me.

. L73: why did you hypothesize this?

. From this point on no more examples of inaccuracies in writing will be given.

. L96-116: some parts of this paragraph are discussing the statistical procedures used. So, those parts should be moved to the Discussion section.

. If the 100 m breaststroke display less competitions than the 200 m why is it given in first place comparing with the “oldest” event?

. It would be fundamental to clearly express what are “tendency to improve“, “substantial differences”, “meaningful difference”, “significant improvements” and “slightly converging tendency”.

. In the figures “time” should be written with capital letter and in the table “1” should be “first” and so on and so forth. Please also replace the commas by points when indicating decimal places.

. there is an evident lack of coherence regarding the length of the paragraphs (some with 20 and others with seven lines).

. When displaying numerical data in a row, the unit should be placed only once (at the final).

. Are any of the reasons for the male vs female differences displayed in the first paragraph of the Discussion swimming specific?

. L285-309: This is a huge text block, being very hard to follow the text flow.

. Discussing the differences between male and female swimmers is a very interesting and actual point of interest in swimming science. However, the authors did not introduce a basic justification why swimmers sex implies different performance – the swimmers economy profile. In fact, it has been studied for some decades the influence of male and female anthropometric characteristics (particularly body fate) on swimmers physiologic and biomechanical profile but the current study does not approach this topic. Authors can find interesting data in the studies of Di Parmpero, Pendergast and Zamparo (in the World book of swimming: From science to performance, you can find ref to studies published in peer-reviewed journals), Chatard, Lavoie and Montpetit (e.g. Eur J Appl Physiol 1990) and Fernandes, Barbosa and Vilas-Boas (e.g. J Hum Movement Stud, 2005).

Reviewer 2 Report

 The article is well written and the authors have done a good job of compiling and presenting the results. From my point of view, this study is enjoyable to read and could be published after some revisions are made. Notwithstanding this work, to me this article presents a major issue that should be improved, justified or fixed, which is: Why have the authors used a linear extrapolation to predict the results? Especially, why have the authors selected the data for this analysis by selecting only from 1972 onwards? I have some doubts about this and about the justification provided by the authors. I understand that if we include pre-1972 data and adopt a linear trend, the prediction would be highly surrealistic, as it would predict much lower results than could actually be achieved by humans, and this does not seem possible, at least for the time being. In this sense, I understand the authors' position that it is logical that human physical-sporting performance has to reach a limit and that therefore, world records will fall much less significantly over the next few years.

However, and with the above mentioned in mind, that is precisely why when looking at the data presented, for example for the butterfly, the trend should have been polynomial, and in my view, the extrapolation of the trend line and therefore the prediction should have been so. Therefore, I do not understand why the authors have not used a polynomial trend, since, in my view, it would be the most appropriate for predicting the outcome based on the justification presented by the authors. I am not saying that this prediction will not come true. We will know in the coming months; however, it is strange to see how the prediction for 2021 is much lower than its predecessors considering that the world records should become more and more difficult to achieve as time goes by. Therefore, I think this may have been a side effect of applying a linear extrapolation.

On the other hand, and if I am not mistaken, I have observed that the authors have used the average of the marks of the Final of each Olympics to establish their trends. I do not mean to say that I see any problem here, but the authors should take into account that the prediction could be different if only the best or worst mark achieved in that championship is taken into account. Perhaps the authors would like to take this into consideration and provide other views in relation to it. Additionally, I have missed some comment or limitation conditioned by the current pandemic situation, or by the fact that athletes have had to modify their preparation plans in order to compete a year later than planned.

Finally, I was curious to see that the predicted best time for the men's 100 breaststroke is even worse than the world record set by Adam Peaty for this event at the World Championships in South Korea in 2019. I understand that the authors have decided to base their analysis solely on the Olympic Games and that is entirely respectable, however, this makes me think that perhaps the authors should have considered other major competitions such as the World Championships, which take place in between the Olympic Games, and which could have given this analysis a much closer representation of reality. Additionally, it would have been interesting to see how the results obtained throughout history could be explained by the interesting social or cultural factors that the authors raise in the discussion. I would have liked to see this in a slightly more specific and not so generic way, with some examples emanating from the results of the paper itself. In fact, it is possible that the authors might decide to revise this section if they eventually decide to adopt another type of data extrapolation technique.

I hope these comments can help to improve the paper.

Good luck.